# Reducing Post-Operative Alveolo-Pleural Fistula by Applying PGA-Sheets (Neoveil) after Major Lung Resection: A Historical Case-Control Study

**DOI:** 10.3390/jcm12072719

**Published:** 2023-04-05

**Authors:** Carolina Sassorossi, Maria Teresa Congedo, Dania Nachira, Diomira Tabacco, Marco Chiappetta, Jessica Evangelista, Adele di Gioia, Velia Di Resta, Claudio Sorino, Michele Mondoni, Fausto Leoncini, Giuseppe Calabrese, Antonio Giulio Napolitano, Adriana Nocera, Achille Lococo, Stefano Margaritora, Filippo Lococo

**Affiliations:** 1Departement of Thoracic Surgery, Università Cattolica del Sacro Cuore, 00168 Rome, Italy; 2Thoracic Surgery, Fondazione Policlinico Universitario A. Gemelli IRCCS, 00168 Rome, Italy; 3Department of Thoracic Surgery, “Pierangeli” Hospital, 65124 Pescara, Italy; 4Division of Pulmonology, Sant’Anna Hospital, 22020 San Fermo della Battaglia, Italy; 5Faculty of Medicine and Surgery, University of Insubria, 21100 Varese, Italy; 6Respiratory Unit, Department of Health Sciences, ASST Santi Paolo e Carlo, San Paolo Hospital, University of Milan, 22020 Milan, Italy; 7Department of Pulmonology, Fondazione Policlinico Universitario A. Gemelli IRCCS, 00168 Rome, Italy

**Keywords:** NSCLC, air-leak, lung cancer, aerostasis, lung resection

## Abstract

Alveolo-pleural fistula remains a serious post-operative complication in lung cancer patients after surgery, which is associated with prolonged hospital stay and higher healthcare costs. The aim of this study is to evaluate the efficacy of a polyglycol acid (PGA)-sheet known as Neoveil in preventing post-operative air-leak in cases of detected intra-operative air-leak after lung resection. Between 11/2021 and 7/2022, a total of 329 non-small cell lung cancer (NSCLC) patients were surgically treated in two institutions. Major lung resections were performed in 251 cases. Among them, 44 patients with significant intra-operative air-leak at surgery were treated by reinforcing staple lines with Neoveil (study group). On the other hand, a historical group (selected by propensity score matched analysis) consisting of 44 lung cancer patients with significant intra-operative air leak treated by methods other than the application of sealant patches were considered as the control group. The presence of prolonged air-leak (primary endpoint), pleural drainage duration, hospital stay, and post-operative complication rates were evaluated. The results showed that prolonged air-leak (>5 days after surgery) was not observed in study group, while this event occurred in four patients (9.1%) in the control group. Additionally, a substantial reduction (despite not statistically significant) in the chest tube removal was noted in the study group with respect to the control group (3.5 vs. 4.5, *p* = 0.189). In addition, a significant decrease in hospital stay (4 vs. 6 days, *p* = 0.045) and a reduction in post-operative complications (2 vs. 10, *p* = 0.015) were observed in the study group when compared with the control group. Therefore, in cases associated with intra-operative air-leak after major lung resection, Neoveil was considered a safer and more effective aerostatic tool and represents a viable option during surgical procedures.

## 1. Introduction

The post-operative alveolo-pleural fistula is one of the most common and insidious complications in thoracic surgery and the main limiting factor for the early discharge of patients from the hospital [1,2,3].

Intra-operative air loss during lung surgery has been reported to occur in 25–75% of patients [4,5], with persistent post-operative air loss (>5 days) occurring in 8–20% of patients [6]. Despite its frequency, there is still no standard approach to manage air leaks, including prolonged ones. However, in post-operative settings different methods, such as observation, continuous or intermittent aspiration, slurry pleurodesis with different agents or surgery revision is frequently used, with each approach having its advantages and disadvantages. Ultimately, the selection of one method over the other is based on the preference of the surgeon.

Due to the lack of a general consensus on the management and treatment of alveolo-pleural fistula, most surgeons would agree that greater effort should be directed towards intra-operative prevention. Moreover, when an air leak is detected at the end of a major lung resection, usual care generally includes direct sutures, restapling, and the use of surgical sealants, which have been recently developed to further reduce air leaks. The most commonly used are liquid fibrin sealants [5,7], synthetic hydrogels [6,8], and collagen fiber bonded sealants [9,10].

In this study, we decided to test the safety and efficacy of a new type of aerostatic product (Neoveil^TM^^®^, Chirurgmed, Pescara, Italy) in a specific subset of lung cancer patients. Therefore, the aim of this study is to evaluate the sealing potency of Neoveil in closing post-operative alveolo-pleural fistula after major parenchymal resections and to determine its effect on the duration of chest drain, the length of patient hospitalization, and post-operative complications [11].

## 2. Materials and Methods

### 2.1. Study Design

This prospective study was conducted in two different thoracic surgery units (Fondazione Policlinico Universitario “A.Gemelli” IRCCS, Università Cattolica del Sacro Cuore, Roma and Private Hospital “L. Pierangeli”, Pescara, Italia). Relevant Ethics Committees approved the protocol (Number 12059/20 ID:3050), and the study was performed in accordance with the Declaration of Helsinki, ICH Good Clinical Practice, and any applicable local regulations. All of the patients provided written informed consent before participating in this study. The study population included patients aged ≥18 years with lung cancer, scheduled for elective pulmonary anatomical resection (segmentectomy, lobectomy, or bilobectomy) with a planned thoracotomy approach or a minimal invasive surgery. Following major pulmonary resections, air leakage was then assessed by a water submersion test under standard airway pressure. The air-leakage intensity was graded by the Macchiarini score as 0 (absent, no apparent leak), 1 (mild, countable bubbles), 2 (moderate, stream of bubbles) or 3 (severe, coalesced bubbles) [12]. However, cases involving the repair of detected air leaks from the bronchial stump (bronchial fistula) were excluded from this report.

Patients with grade 2 or more air leakage were treated with Neoveil alone or after routinely aerostatic methods (redo manual or mechanical sutures). The study group (Group A) was compared with a historical cohort of patients (Group B) whose data had already been prospectively collected in the past, with similar characteristics treated by routinely aerostatic methods only. In Group B, we excluded those cases where any sealant had been used as an aerostatic method. We analyzed this clinical information: age, gender, BMI, smoking habits, FEV1% preoperative, FVC, Dlco, type of lung resection, type of surgical approach used (open vs. VATS), presence of pleural adhesions, characteristics of the fissure (“completely fused“, ”partially fused“, or ”free”), duration of surgery (min), histology, TNM pathological staging (TNM VIII Edition), intra- and post-operative complications, post-operative air leak (total volume of air leak), daily count of drained fluid and air leak (digital detection), the need for suction drainage at negative pressures, the duration of drainage (days), post-operative hospitalization, and the readmission of the patient with emphysema and/or pneumothorax within 30 days after the procedure. Vital signs and clinical laboratory measurements were also reported. In addition, during the study period, we measured the air loss digitally using the REDAX Drentech Palm system, which has been in use for decades in both centers.

The drain was left in a water valve (2 cm H_2_O), and only in the case of the post-operative onset of pneumothorax or subcutaneous emphysema was a negative suction of −20 cm H_2_O applied. The drainage was removed 24 h after the amount of drainage did not exceed 250 cc of serous fluid and after any sign of air leak subsided.

All of the data collected from the two centers were anonymized and entered into a single database before the final analysis to ensure the privacy of the study participants.

### 2.2. Material and Methods

Neoveil is a porous fibrous structure (100% PGA) that easily adheres to the lung parenchyma on the stapler suture line (Figure 1).

In vivo studies involving the use of PGA sheets indicate that the bio-absorption process should be complete after 2–4 weeks [13,13]. The sheet is normally placed on the injured lung surface (where air-leak is detected at lung re-inflation) with a few drops of water, which takes only a few minutes to be applied. Air leaks in the study group were treated with the application of the Neoveil patch on sutured lung parenchyma, after which the air leak was evaluated by the anesthesiologist through a specific sensor in the ventilation circuit.

### 2.3. Methodology and Statistics

Categorical variables were reported as numbers (%), while continuous variables were expressed as mean ± standard deviation (SD). Normal distribution of data was estimated by the Kolmogorov–Smirnov test. Categorical variables were compared by Chi-square test and continuous variables by an independent sample Student’s *t*-test.

Given the comparison with a historical cohort (group B), the two groups were not randomized. To overcome this selection bias, a 1:1 propensity score using the nearest neighbor matching method was first performed to balance the baseline characteristics of the two groups. Variables included in the propensity score model were those that might have influenced clinical decision with the aerostatic methods: age, gender, smoking habits, COPD, cardiovascular diseases, forced expiratory volume in one second, forced vital capacity, type of fissure, adhesions, etc.

After 1:1 propensity score matching, 44 patients were eligible in each group. A *p*-value of less than 0.05 was considered statistically significant. Statistical analysis was performed using IBM SPSS Statistics for Macintosh, Version 25.00 (Armonk, NY, USA). The presence of prolonged air-leak (primary endpoint), and the duration of pleural drainage, hospital stay, and post-operative complication rates, were evaluated.

## 3. Results

Between November 2021 and July 2022, a total of 329 NSCLC patients were surgically treated in the two institutions. Among them, 251 were major lung resections, and at surgery, a total of 44 patients met the inclusion criteria and were enrolled in this prospective study. As reported above in detail, we compared the study group with a control group that underwent standard surgical treatment (suture, restapling, or no further treatment) after the same anatomical lung resection. After 1:1 propensity score matching, the baseline clinicopathologic characteristics did not differ between these two groups.

In the study group, out of the 44 patients, 30 were men with an average age of 66 ± 5.3 years, while in group B there were 28 males (average age = 64 ± 5.3 years). A smoking habit was reported in 43% of patients, while 45% were presented with COPD in both groups. Lobectomy was the most common surgical procedure performed in both groups (see Table 1).

At surgery, in the study group, 14 patients had complete fissure, while 30 patients had partially incomplete or totally incomplete fissure, which were reported in the surgical notes. In contrast, in the control group, complete fissure was detected in 15, while 27 patients were presented with fully fused fissure at surgery.

In addition, pleural adhesiolysis was performed in about one third of the cases in both the study group and the control group, with 13/44 patients and 12/44 patients, respectively.

Concerning the primary end-point of the study, we did not observe any prolonged air-leak (>5 days after surgery) in the study group, while this event occurred in four patients (9.1%) in the control-group (*p* = 0.041) (Table 2).

The treatment of persistent air-leaks (observed in 4 cases) was conservative (“wait and see” strategy) in 3 of them with a drainage mean duration of 11 days (from surgery). In the remaining case, a re-operation (re-do suture of the lung parenchyma) was necessary for a persistent air-leak at the 10th post-operative day. As a consequence of this, we observed a remarkable (despite not statistically significant) reduction in pleural drainage duration (3.5 vs. 4.5, *p* = 0.189) and a significant decrease in the hospital stay (4 vs. 6 days, *p* = 0.045). We also observed, in the study group, a reduction in perioperative complications (2 vs. 10, *p* = 0.015).

## 4. Discussion

A polyglycolic acid mesh sheet (Neoveil™) is a tissue-strengthening repair agent that prevents air or fluid leakage after surgery. The preventive effect of Neoveil has been proven in various surgical fields; these sheets reduce the pancreatic fistula incidence in pancreas resection [14,15,16,17], prevent bile leakage and hemorrhage after liver resection [18], and prevent bowel content leakage after colon surgery [19].

In lung surgery, Neoveil has been successfully adopted to treat post-operative bronchial stump fistula after lobectomy [20,21] or to cover a bulla and reduce the recurrence of a pneumothorax [22]. More recently, Miyahara and co-workers reported its efficacy to reduce the recurrence rate after bullectomy for primary pneumothorax [23]. In detail, by comparing two groups of patients who underwent bullectomy alone vs. bullectomy with Neoveil sheets used to cover the visceral pleura, they observed a significantly lower recurrence rate in the Neoveil group (2.6% vs. 24.8%, *p* < 0.000001).

In this study, we selected lung cancer patients with intra-operative significant alveolo-pleural fistula (air-leakage detected at the end of surgical maneuvers) to test the efficacy of Neoveil sheets on reducing the occurrence of persistent air-leak (>5 days after surgery).

The choice of these subset of patients (study group) is in line with a recent consensus of experts reported by Brunelli and co-workers [24]. By assessing the clinical practices of a large audience of European thoracic surgeons, the authors suggested that for the correct use of sealants, all of the patients at high-risk of post-operative air leak before surgery (i.e., COPD, emphysema, etc.) should not be included. However, they recommended the selection of patients with a significant intra-operative air-leak (regardless of other variables) for the application of sealants.

According to our results, the Neoveil patch seems to be safe and efficient at reducing post-operative alveolo-pleural fistula in patients who underwent major lung resection for lung cancer. In particular, among 44 patients with a significant intra-operative air leak, we did not observe any persistent air leak 5 days after surgery when applying the Neoveil sheet at surgery. In a similar population (historical control group), about 10% of cases reported having a persistent air leak, and this percentage corresponded with other similar experiences in this field [4,5,6]. In fact, while intra-operative alveolo-pleural fistula are very common complications after lung resection for lung cancer, ranging from 25 to 75% [4,5] of cases, the persistence of air leak more than 5 days after surgery is less frequent (reported in about 8% to 20% of cases) but is still considered as the first cause of prolonged hospital stay [6]. Various surgical centers have reported that alveolo-pleural fistula increases post-operative morbidity and mortality, with a lengthened hospital stay [25,26,27,28] and increased incidence of empyema [29] as the main consequences.

From our analysis, we have observed that, as a consequence of a reduction post-operative air leak, the hospital stay and the complication rates were significantly lower in the study group where Neoveil had been used compared with the historical control group (hospital stay: 4 vs. 6, *p* = 0.045; post-op complication: 4.7% vs. 22.7%, *p* = 0.015).

A recent meta-analysis performed by McGuire and colleagues [30] on this topic seems to support our findings in terms of reduced hospital stay. Specifically, the authors selected 44 articles from a systematic research (from 1948 to 2018), with a total of 2537 cases. In particular, 1292 of them were randomized to an intervention group of pulmonary resections with an intra-operative application of a polymeric sealant, while 1245 were assigned to the control group of pulmonary resection with aerostasis by sutures.

Despite inter-trial heterogeneity, the authors found out that the duration of air leak, chest tube management, and length of hospital stay were all reduced by an average of one day with the use of polymeric sealants [30].

The choice of the best sealant is also a very debated issue. In the aforementioned consensus, the thoracic surgeons that participated in the survey discussed how to determine the “ideal sealant”. According to them, the “ideal sealant” should be flexible enough to allow its application on a partially inflated lung and its specific use in thoracic surgery (i.e., long application). Additionally, it should be re-absorbable, provide adequate adherence to lung surfaces, and not take more than five minutes to be applied. Ultimately, Neoveil was found to possess most of these characteristics; hence, it is perceived to be safer and more efficient than other sealants, after a major lung resection.

## 5. Limitations and Points of Strengths of the Study

This study has several limitations that should be taken into account when interpreting these results. Firstly, we selected a historical control over a long period (about 3 years) based on a propensity score match analysis. Although the statistical method used for analysis is considered appropriate for case-control studies (variables are equally distributed between groups compared each other), a randomized trial would have been better to test the efficacy of a medical device. In addition, the number of patients enrolled in this study is considered too small to make definite conclusions. Overall, these biases may have affected the results of this study. Consequently, these results should be interpreted with caution.

Despite these limitations, this study has the merit of focusing on a specific population of NSCLC cases with significant intra-operative air-leak, collecting a prospective data set with homogeneous therapeutic algorithms (shared by the two institutions).

## 6. Conclusions

Based on the results from this study, Neoveil was proven to be safe, effective, and useful in reducing post-operative alveolo-pleural fistula in patients with intra-operative air-leak during elective pulmonary anatomical resection. Readers should interpret these results, even if coming from a small group study, with caution as they may further lead to the development of concepts in relation to enhanced recovery after surgery, especially in cases of intra-operative air leak after major lung resections, thereby improving the patients’ quality of life.

## Figures and Tables

**Figure 1 jcm-12-02719-f001:**
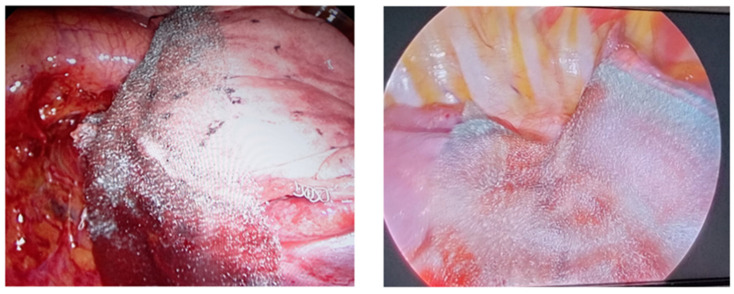
Neoveil sheets application on lung surface.

**Table 1 jcm-12-02719-t001:** Distribution of the main clinical and surgical characteristics between the Group A and Group B.

	Group A (%)	Group B (%)	*p*-Value
Gender (male)	30 (68.2%)	28 (63.6%)	0.653
Age	66.64 ± 8.93	64.23 ± 11.06	0.267
Smoking	19 (43.2%)	18 (40.9%)	0.901
COPD	21 (47.7%)	19 (43.2%)	0.817
Pre-operative FEV1%	95.09 ± 16.05	82.26 ± 45.74	0.083
DLCO	24.60 ± 5.57	23.77 ± 6.21	0.675
Type of fissure			
*Complete*	14 (31.8%)	15 (34.1%)	0.135
*Partially complete*	4 (9.1%)	0	
*Incomplete*	26 (59.1%)	27 (61.4%)	
Pleural adhesions	13 (29.5%)	12 (27.3%)	0.866
Type of resection (Lobectomy)	37 (84.1%)	39 (88.6%)	0.810

**Table 2 jcm-12-02719-t002:** Primary and secondary end point.

	Group A (%)	Group B (%)	*p*-Value
Air-leakage >5 days	0	4 (9.1%)	**0.041**
Duration of pleural drainage	3.5	4.5	0.189
Duration of hospital stay	4	6	**0.045**
Post operative complication	2 (4.7%)	10 (22.7%)	**0.015**

Bold values indicate statistical significance.

## Data Availability

Data are the property of the Fondazione Policlinico Universitario A. Gemelli and may be visualized if needed after authors approval.

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
