# Peer review of "Reducing Post-Operative Alveolo-Pleural Fistula by Applying PGA-Sheets (Neoveil) after Major Lung Resection: A Historical Case-Control Study"

_jcm, 2023, doi:10.3390/jcm12072719_

Round 1

Reviewer 1 Report

First, Thank you for this work. 

In this article authors have evaluated the PGA-sheet sealing Neoveil (PGA-N) for the post operative prevention and reduction of air-leaks.

            It’s a very interesting topic because, prolonged air leaks are a major issue in thoracic surgery, leading to an increased risk of post operative pneumopathy and also a longer hospital length of stay and an increased healthcare costs. Moreover, with the increase in infralobar resections this complication is more prevalent.

            Concerning the methodology:

-        Congratulation for this prospective trial. Not easy to build and to manage, moreover in 2 institutions.

-       Concerning the evaluation scale of air leakage, it’s a debated subject. Macchiarini score is still debated, why not using the ratio of measured air leaked in the mechanical respiratory? Moreover this kind of test is difficult to manage in minimally invasive approach and not always reproducible. But there isn’t a standardized and recommended approach to measure post operative air leak. 

-       Concerning the statistical analysis, I have no specific comments, methodology and tests seems to be the good ones. 

-       Authors made a choice by the selection of Macchiarini score of 2 according recommendations but that is also debated. Groups seems to be representative from a clinical point of view. This suggests a common practice. Nevertheless, we regret that there were only 44 patients, and the period could have been longer to recruit more patients.

-       Concerning the type of resection, I regret the few proportion of segmentectomy where this issue is a major issue and challenge.

-       A video of the way of scaling air leak and the way of applicating PGA-N is welcome. 

Concerning the results:

-       Results are interesting, but the statistical significance remains low and borderline. Interpretation must be cautious.

-       Do you have any data about the medico-economic evaluation of this device and the reduction of healthcare costs generated? 

Concerning the discussion and the results:

-       Results are debated with cautions, congratulation. 

-       As it’s mentioned the “miracle air sealant” as not been developed! 

            As a conclusion it’s an interesting article, well written, with cautions conclusions. Nevertheless the air leak scale could be debated, and maybe more patients could be analyzed. But congratulation for this prospective trial and for your caution.

Your next work is to focus on post operative air leak after segmentectomy with is a a major issue. 

Nevertheless, congratulations to authors for this article. 

Author Response

We are grateful to the reviewers for their relevant comments and the work they have invested to help us improving our manuscript. We believe that their positive remarks and questions allowed us to increase the interest of our manuscript.

Please find below the responses to reviewers’ comments.

Reviewer 1, Round #1 Comments for the Author:

  • First, Thank you for this work.

In this article authors have evaluated the PGA-sheet sealing Neoveil (PGA-N) for the post operative prevention and reduction of air-leaks.

            It’s a very interesting topic because, prolonged air leaks are a major issue in thoracic surgery, leading to an increased risk of post operative pneumopathy and also a longer hospital length of stay and an increased healthcare costs. Moreover, with the increase in infralobar resections this complication is more prevalent.

            Concerning the methodology:

-        Congratulation for this prospective trial. Not easy to build and to manage, moreover in 2 institutions.

Reply: Thanks for your kind words. We have appreciated the fact that You understood our effort to build and manage such type of trial study. Thanks again.

  • Concerning the evaluation scale of air leakage, it’s a debated subject. Macchiarini score is still debated, why not using the ratio of measured air leaked in the mechanical respiratory? Moreover this kind of test is difficult to manage in minimally invasive approach and not always reproducible. But there isn’t a standardized and recommended approach to measure post operative air leak.

Reply: Thanks very much for your insightful and constructive comment. Actually, as correctly remarked, at today there isn’t a standardized and recommended approach to measure post operative air leak. Macchiarini’s score is the usual methods in our Institution but in the last years we have tried to adopt the measured air leaked in the mechanical respiratory in our clinical practice. The issue is that there is no threshold or a cut-off already validated in literature of the intra-op air leaks measured by the mechanical respiratory to use to clinical trial. On the other hand, Macchiarini’s score, despite old and probably surpassed has been adopted in a large literature experience and this is why we decided to use it as “score” to enroll patient in the present trial. However, as You may observe in the manuscript text (see last sentence of paragraph 2.2.), we also adopted the air-leak measurement by the mechanical respiratory before closing the thorax. In our daily clinical practice, we consider a good aerostatic result if we observ a final intra-op air-leaks <50 ml per respiratory act with a standard Tidal volume (about 300-350ml per respiratory act) and a standard level of positive end-expiratory positive pressure (PEEP = 2-3 cmH2O). Finally, at the beginning of our learning curve in uniportal VATS (2014), the evaluation of air-leak was almost difficult and challenging but at today (about 8 years later) we have adopted few tricks that make this evaluation easily feasible, thanks also to a nice cooperation with our anesthesiologists.

  • Concerning the statistical analysis, I have no specific comments, methodology and tests seems to be the good ones.

Reply: Thank you for your nice consideration.

  • Authors made a choice by the selection of Macchiarini score of 2 according recommendations but that is also debated. Groups seems to be representative from a clinical point of view. This suggests a common practice. Nevertheless, we regret that there were only 44 patients, and the period could have been longer to recruit more patients.

Reply: Thank you for your proper consideration. We decided to not include a large dataset of patients with pre-operative factors of risk for air-leaks (as previously done by other colleagues) but to focus our attention on those patients with an intra-operative remarkable air-leak only. Moreover, we included only lung cancer patients and only mayor resections in order to increase the homogeneity of the study sample. Accordingly, our dataset consisted of a highly selected cohort and on it we calculated the minimum sample size to perform a correct statistical analysis. By the way, we agree with You that, clinical practice is different from clinical trial. Indeed, we are continuing to enroll patients in the study group (Neoveil) and we expect to reach the first 100 cases in the next months in order to have more robust results to publish on this topic.

  • Concerning the type of resection, I regret the few proportion of segmentectomy where this issue is a major issue and challenge.

Reply: This is a very interesting observation; this trial was ideated and led a couple oy years ago and, unfortunately, before the recent publication of the study JCOG0802, that, as you know, have finally demonstrated the non inferiority of segmentectomy compared to lobectomy for tumor smaller than 2 cm and with a consolidation-to-tumor ratio>0.5. Our policy has largely changed in the last months and, at today we currently perform segmentectomy in early-stage lung cancer by uniportal VATS, but these data were antecedent to the study period. In the next analysis we will  have data coming from segmentectomy to better understand the role of Neoveil for air leak in these cases.

  • A video of the way of scaling air leak and the way of applicating PGA-N is welcome.

Reply: Thanks for your comment. We have different videos of NEOVEIL application and, in fact the photos reported in the text are just a screenshot of one of them. These materials are at your disposal but we are not allowed to include one video in this step of the manuscript submission process. Please send your email to filippo.lococo@policlinicogemelli.it and we will send it.

Alternatively, I will ask to the Editor to include it in the manuscript.

  • Results are interesting, but the statistical significance remains low and borderline. Interpretation must be cautious.

Reply: Thank you for this observation. As stated before, our dataset consisted of a highly selected cohort and on it we calculated the minimum sample size to perform a correct statistical analysis. Despite this, we agree with you on the fact that the number of patients per group is quite low and, inspired by your comment we have decided to further underline in the text the fact the results have to be interpreted very cautiously by the Readers (see Discussion and Conclusion sections).

  • Do you have any data about the medico-economic evaluation of this device and the reduction of healthcare costs generated?

Reply: We are positively impressed by your comment that will give us the possibility to explore a further and interesting aspect of this issue. As You correctly guessed, the reduction of persistent post-op air-leaks may significantly increase the length of hospital stay, this resulting in higher healthcare costs. From a preliminary raw analysis of these data performed by our Institutional Staff, the reduction of healthcare costs is significant even considering the costs of the aerostathic device (Neoveil). We will perform a more comprehensive and complete cost-effective analysis in the next manuscript (as said before)

  • Concerning the discussion and the results:

Results are debated with cautions, congratulation.

As it’s mentioned the “miracle air sealant” as not been developed! As a conclusion it’s an interesting article, well written, with cautions conclusions. Nevertheless the air leak scale could be debated, and maybe more patients could be analyzed. But congratulation for this prospective trial and for your caution.

Your next work is to focus on post operative air leak after segmentectomy with is a a major issue.

Nevertheless, congratulations to authors for this article

Reply: We appreciate your nice observations. We concur that we are still defining an accurate method to evaluate and standardize the intra-operative air-leak measurement and that the ideal sealant does not exist at today. Even considering all the limitations that You correctly remarked, we hope these preliminary results may be of help the physicians involved in this field.

On the behalf of all co-Authors,

Carolina Sassorossi

Reviewer 2 Report

Thank you for giving me this kind of precious opportunity to review this manuscpript.  This paper deals with the reduction of alveolo-pleural fistula after lung resection using PGA sheet.   On the other hand, this study area and methods can be overlapped previous studies and I think that this paper has some problems as indicated below.   1.  The authors should define the abbreviations. What is the meaning of "PNX"?   2.The manuscript such as below sentences should be improved by a thorough English language review  before acceptance for publication.   Page 3. Lines 8-9 "few" should be changed to "a few" Page 4, Line 14 A  "adesiolisis" should be changed to "adhesiolysis".   3.  The subtitle and content should be changed. 2.3. "Methods" is "Statistics"?   4. The author described that "Drainage mean duration of 11 days (from surgery) ". However, duration of pleural drainage is 3.5 days and 4.5 days respectively in Table 2. These are inconsistent. The authors should explain it.   5. Were leak points sutured using needle thread or stapler? I think that NEOVEIL can't bond the lung immediately each other without fibrin glue. Major leakage such as cases of COPD can not be stoped using NEOVEIL only. What the authors think about? 

Author Response

Reviewer 2, Round #1Comments for the Author.

  • Thank you for giving me this kind of precious opportunity to review this manuscpript.  This paper deals with the reduction of alveolo-pleural fistula after lung resection using PGA sheet. On the other hand, this study area and methods can be overlapped previous studies and I think that this paper has some problems as indicated below. 

Reply: Thank you for your observation; firstly, we agree with You on the fact that this topic has already been faced by many other author. However, we think that the originality of this study consisted of the selection patients. In details, we decided to not include a large dataset of patients with pre-operative factors of risk for air-leaks (as previously done by other colleagues) but to focus our attention on those patients with an intra-operative remarkable air-leak only. Moreover, we included lung cancer patients and mayor resections only in order to increase the homogeneity of the study sample. Accordingly, our dataset consisted of a highly selected cohort of patients to be evaluated. Finally, we have tried to take into account the Reviewers’ comment in order to edit the manuscript and increase its overall quality. Thus, we have largely appreciated the time You have invested to review the present manuscript.

  • The authors should define the abbreviations. What is the meaning of "PNX"?  

Reply: Thanks. We apologize for this wrong use of the acronym in the text. The term PNX means pneumothorax and we have corrected it on the revised version of the manuscript.

  • The manuscript such as below sentences should be improved by a thorough English language review before acceptance for publication.  

Page 3. Lines 8-9 "few" should be changed to "a few"

Page 4, Line 14 A  "adesiolisis" should be changed to "adhesiolysis".  

Reply: The overall manuscript has been reviewed by a native English colleague of our Institution (Dr. Diepriye Charles-Davies) and he has extensively the English language and made all the necessary corrections (see Track version of the revised manuscript).

  • The subtitle and content should be changed.

Reply: We apologize for not understanding this request, since we have no subtitle. Please clarify your point. We are at Your disposal.

  • "Methods" is "Statistics"?  

Reply: Thanks for such correct observation. We change the title of the paragraph  “Methodology and statistics”, as suggested.

  • The author described that "Drainage mean duration of 11 days (from surgery) ". However, duration of pleural drainage is 3.5 days and 4.5 days respectively in Table 2. These are inconsistent.  The authors should explain it.  

Reply: Thank you for your right observation, and we sincerely apologize because this sentence was effectively confounding. In text we talked about 4 cases of prolonged air leak (>5 days) and not about the overall group of 44 cases. When evaluating these 4 cases only, in 3 patients where a wait and see strategy was applied, we observed a mean chest tube duration of 11 days, this explaining the higher chest tube duration of the overall group (4.5 days) compared to the Neoveil group (3.5 days).  

We have clarified this concept in the text and thanks again for your revision.

  • Were leak points sutured using needle thread or stapler? I think that NEOVEIL can't bond the lung immediately each other without fibrin glue. Major leakage such as cases of COPD can not be stopped using NEOVEIL only. What the authors think about?

Reply: Thanks for your proper comment that give us the opportunity to discuss in deep this aspect. As correctly stated by You, Neoveil differently by the glues does not act immediately. We generally observed only a minimal reduction of the air-leak after applying the Neoveil sheets as a consequence of a mechanical effect of them. However, thanks to the digital evaluation of the post-op daily air-leak, we observed a strong reduction of the air-leak between the 24th and the 76th post-operative hours. This observation suggest that this sealant  needs some time to be incorporated along the lung surface. As correctly remarked and in line with our practical experience, a severe intra-op air-leak in COPD patients deserves the use of stapler, but these cases even after stapler a remarkable air-leak may be detected and Neoveil could be of help.

To better clarify this aspect we have re-wrote the methodology as reported herein:

« Patients with grade 2 or more air leakage were treated with Neoveil alone or after routinely aerostatic methods (redo manual or mechanical sutures). The study group (Group A) was compared with a historical cohort of patients (Group B), whose data had already been prospectively collected in the past, with similar characteristics treated by routinely aerostatic methods only. In Group B we excluded those cases where any sealant were used as aerostatic method ». We hope that this version was more clear for the Readers.

On Behalf of all co-Authors

Carolina Sassorossi

Round 2

Reviewer 2 Report

Thank you for your reply to my questions point by point. It is clear to see that a lot of hard work has been put into the study. This paper is an important contribution and I recommend that it be accepted for publication.